# Maternal Dietary Selenium Intake during Pregnancy Is Associated with Higher Birth Weight and Lower Risk of Small for Gestational Age Births in the Norwegian Mother, Father and Child Cohort Study

**DOI:** 10.3390/nu13010023

**Published:** 2020-12-23

**Authors:** Pol Solé-Navais, Anne Lise Brantsæter, Ida Henriette Caspersen, Thomas Lundh, Louis J. Muglia, Helle Margrete Meltzer, Ge Zhang, Bo Jacobsson, Verena Sengpiel, Malin Barman

**Affiliations:** 1Department of Obstetrics and Gynaecology, Sahlgrenska Academy, University of Gothenburg, 405 30 Gothenburg, Sweden; pol.sole.navais@gu.se (P.S.-N.); bo.jacobsson@obgyn.gu.se (B.J.); verena.sengpiel@obgyn.gu.se (V.S.); 2Division of Infection Control, Environment and Health, Norwegian Institute of Public Health, 0213 Oslo, Norway; annelise.brantsaeter@fhi.no (A.L.B.); hellemargrete.meltzer@fhi.no (H.M.M.); 3Centre for Fertility and Health, Norwegian Institute of Public Health, 0403 Oslo, Norway; ida.henriette.caspersen@fhi.no; 4Department of Laboratory Medicine, Division of Occupational and Environmental Medicine, Lund University, 221 85 Lund, Sweden; thomas.lundh@med.lu.se; 5Cincinnati Children’s Hospital Medical Center, Division of Human Genetics and Perinatal Institute, Cincinnati, OH 45229, USA; Louis.Muglia@cchmc.org (L.J.M.); Ge.Zhang@cchmc.org (G.Z.); 6Department of Pediatrics, University of Cincinnati College of Medicine, Cincinnati, OH 45267, USA; 7Office of the President, Burroughs Wellcome Fund, Research Triangle Park, Durham, NC 27709, USA; 8Department of Genetics and Bioinformatics, Domain of Health Data and Digitalisation, Institute of Public Health, 0456 Oslo, Norway; 9Department of Obstetrics and Gynaecology, Sahlgrenska University Hospital/Östra, 405 30 Gothenburg, Sweden; 10Department of Biology and Biological Engineering, Chalmers University of Technology, 412 96 Gothenburg, Sweden; 11Institute of Environmental Medicine, Karolinska Institutet, 171 77 Stockholm, Sweden

**Keywords:** birth weight, intrauterine growth, selenium, the Norwegian Mother, Father and Child Cohort study, MoBa, Medical Birth Registry of Norway, MBRN

## Abstract

Selenium is an essential trace element involved in the body’s redox reactions. Low selenium intake during pregnancy has been associated with low birth weight and an increased risk of children being born small for gestational age (SGA). Based on data from the Norwegian Mother, Father and Child Cohort Study (MoBa) and the Medical Birth Registry of Norway (MBRN), we studied the association of maternal selenium intake from diet and supplements during the first half of pregnancy (*n* = 71,728 women) and selenium status in mid-pregnancy (*n* = 2628 women) with birth weight and SGA status, according to population-based, ultrasound-based and customized growth standards. An increase of one standard deviation of maternal dietary selenium intake was associated with increased birth weight z-scores (*ß* = 0.027, 95% CI: 0.007, 0.041) and lower SGA risk (OR = 0.91, 95% CI 0.86, 0.97) after adjusting for confounders. Maternal organic and inorganic selenium intake from supplements as well as whole blood selenium concentration were not associated with birth weight or SGA. Our results suggest that a maternal diet rich in selenium during pregnancy may be beneficial for foetal growth. However, the effect estimates were small and further studies are needed to elucidate the potential impact of selenium on foetal growth.

## 1. Introduction

Intrauterine growth retardation (IUGR) and small for gestational age (SGA) are associated with intrauterine foetal death [1], cerebral palsy [2], perinatal acidosis, hypoglycaemia, hypothermia, and coagulation abnormalities [3]. Later in life, SGA born children are at higher risk of short stature, cognitive delays, diabetes and cardiovascular disease [3,4,5,6]. Some risk factors for having a baby born SGA, such as maternal smoking, maternal weight (very low and also increased body mass index) and primiparity, have been identified [7,8,9,10]. In addition, a number of dietary factors have been associated with SGA, such as diet quality/unhealthy dietary patterns [11,12,13], high sugar consumption in pregnancy [14], low consumption of seafood [15,16], low iodine intake [17], and caffeine consumption [18,19]. A review on the impact of maternal diet during pregnancy on infant birth weight found that consumption of whole foods such as fruit, vegetables, low-fat dairy, and lean meats throughout pregnancy may be beneficial for appropriate birth weight in relation to gestational age [20]. Studies examining associations between maternal/neonatal trace element concentrations during pregnancy and SGA status found selenium concentration, but not iron, zinc, copper, cadmium or lead, to be significantly associated with the risk of being born SGA [21,22,23].

Selenium is a trace element which is incorporated into selenoproteins. Examples of important selenoproteins are the key antioxidative enzyme glutathione peroxidase and the iodothyronine deiodinases D1, D2, and D3, each containing selenium at their active sites [24]. Glutathione peroxidase has a major impact on redox status and regulates oxidative stress, while the iodothyronine deiodinases have key roles in regulating circulating and intracellular levels of thyroid hormones [25]. We have previously reported that maternal dietary selenium intake is associated with a small but significant decrease in the risk of preterm delivery and an increase in gestational duration [26]. There are also studies suggesting that maternal blood selenium status before or during pregnancy correlates positively with birth weight [23,27].

The aim of this study was to examine whether maternal intake of selenium and maternal whole blood selenium concentration in mid-pregnancy are associated with birth weight and SGA in a large Norwegian pregnancy cohort.

## 2. Materials and Methods

### 2.1. Study Population

The Norwegian Mother, Father and Child Cohort Study (MoBa) is an ongoing, prospective, population-based pregnancy cohort administered by the Norwegian Institute of Public Health [28] including women from all of Norway between 1999 and 2008. Pregnant women were invited by postal invitation in connection with the routine ultrasound screening in gestational week 18. The women consented to participation in 41% of the pregnancies. The cohort now includes 114,500 children, 95,200 mothers and 75,200 fathers [28]. Participants were asked to fill in questionnaires covering a wide range of information at several time points during pregnancy and childhood. This study is based on information from questionnaire 1 (Q1) about general health status and lifestyle filled in around gestational week 15 to 17 and on the semi-quantitative food frequency questionnaire (FFQ) filled out around gestational week 22 (Q2). Pregnancy and birth records from the Medical Birth Registry of Norway (MBRN) were linked to the MoBa dataset [28]. MBRN is a national health registry containing information about all births in Norway. Blood samples were obtained from both parents during pregnancy and from mothers and children (umbilical cord) at birth. The current study is based on version 10 of the quality-assured data files released for research in 2017. The establishment and initial data collection in MoBa were based on a license from the Norwegian Data Protection Agency and approval from The Regional Committee for Medical and Health Research Ethics. The MoBa cohort is now based on regulations related to the Norwegian Health Registry Act. The current study was approved by The Regional Committee for Medical Research (2015/2425/Rek South-East A).

Only women with singleton live births after gestational week 22^+0^ and women with valid information on selenium intake (i.e., total energy intake between 4.5 and < 20 MJ/day and less than 3 blank pages in the FFQ) were included. Women who participated in the cohort with more than one pregnancy were included in the present study only with their first enrolled pregnancy (see Figure 1 for selection of the study population). Subjects with > 4 standard deviations from the mean in the birth weight or selenium intake variables were excluded, resulting in 71,728 individuals included in the statistical analyses for selenium intake and 2628 for analysis of selenium status (Figure 1).

### 2.2. Maternal Intake of Selenium

Maternal daily intake (µg/day) of selenium from food and dietary supplements was estimated based on self-reported intake using the semi-quantitative MoBa food frequency questionnaire (FFQ) designed to assess habitual diet and intake of dietary supplements during the first four to five months of pregnancy [29]. The FFQ included questions about the intake of 255 food items or dishes and the FFQ was used from March 2002 throughout the remaining recruitment period. The questionnaire was optically read, and the consumption frequencies were converted into food amounts (gram/day) using standard Norwegian portion sizes. FoodCalc [30] and the Norwegian food composition table were used for calculating the intakes of selenium and other nutrients from food. As previously described [26], the participants reported the use of dietary supplements by writing the name and brand of the supplement as well as the frequency and amount. Selenium intake from supplements was calculated from a database of more than 1000 dietary supplements [31]. The content of selenium in the different supplements was based on the producers’ declared nutrient content information. The selenium supplements contained one or more forms of selenium, including inorganic selenite or selenate and organic forms of selenium such as selenomethionine and selenised yeast. The supplementary intake of selenium was analysed separately for organic and inorganic supplements, since these forms differ in impact on tissue selenium concentration.

The FFQ used in MoBa has been thoroughly validated in 119 MoBa participants using a four-day weighed food diary and biological markers in blood and urine as reference measures [32,33]. The validation study showed that the MoBa FFQ is a valid tool for assessing dietary intake of energy, nutrients and food in the first four to five months of pregnancy.

### 2.3. Selenium Concentration in Whole Blood

Data on selenium concentration in whole blood were available for a sub-sample of 2638 women included in the Norwegian Environmental Biobank [34,35]. Whole blood collected in heparin tubes in gestational weeks 17–18 was shipped by ordinary mail (unrefrigerated shipment) in a vacutainer for long-term storage at −20 °C at a central biorepository [35,36]. Selenium was analysed at Lund University, Sweden, by inductive couple plasma mass spectrometry (ICP-MS; iCAP Q, Thermo Fisher Scientific, Bremen, Germany, GmbH) equipped with a collision cell with kinetic energy discrimination and helium as collision gas. The detection limit was 3.2 µg/L and the coefficient of variation was 1.5%. All samples were above the detection limit. The analytical accuracy was verified against certified reference material, Seronorm Trace elements whole blood L-1 and L-2 (SERO AS, Billingstad, Norway).

### 2.4. Birth Weight and SGA

Birth weight was recorded in the MBRN [37]. In addition to birthweight in grams, birth weight was examined as continuous (z-scores) and dichotomous (small for gestational age, SGA) outcome variables based on the following growth standards:(a)Population-based SGA:<10th percentile according to MBRN birth weight z-score based on infants’ weight gestational age sex from all deliveries included in MBRN from 1967 to present. Calculated by MBRN based on Skjærven’s gestational age-based growth curves [38].(b)Ultrasound-based SGA: >2 standard deviations (SD) below the expected birth weight (z-score) for any given gestational age according to Marsal’s ultrasound-derived growth curves [39].(c)Customized SGA: <10th percentile of birth weight z-score according to Gardosi et al. [40] based on Hadlock et al.’s ultrasound-derived growth curves [41], and taking infant sex, maternal weight, height, and parity into account [42].

The population-based definition of SGA (a) is used as the main outcome and these results are presented in the main text, while the results from the ultrasound-based (b) and the customized-based (c) SGA analyses are presented in the Appendix A).

### 2.5. Covariates

Multivariate analyses were performed with and without adjustment for the following potential confounders: maternal age at delivery, maternal pre-pregnancy body mass index (BMI), parity, maternal smoking habits during pregnancy, passive smoking, and maternal education. Models for selenium intake (from diet or from inorganic or organic supplements) were also adjusted for nausea during second trimester, fibre intake, iodine intake, protein intake, n-3 intake from diet and total energy intake, as well as mutually adjusted for the different selenium sources (diet intake, organic supplements and inorganic supplements).

Fish intake has previously been found to be positively associated with higher birth weight in MoBa [43]. To account for confounding from other compounds in fish, all models were adjusted for marine omega-3 polyunsaturated fatty acids from food. Other important sources of selenium in the diet are meat and dairy products. Protein intake from dairy products has been found to be associated with birth weight and SGA [44]. To account for confounding by meat and dairy products, total protein intake was included in all adjusted models. Iodine was included in the model for dietary selenium since selenium and iodine share some of the same food sources and iodine has been found to be associated with foetal growth in MoBa [17]. In addition, we adjusted for total energy intake to account for an overall higher intake among women with high selenium intake and for fibre intake to account for an overall healthy eating behaviour.

Data on maternal age at delivery were available from MBRN and used as a continuous variable. Maternal pre-pregnancy BMI was based on self-reported pre-pregnancy height and weight and grouped according to the WHO classification as underweight (<18.5 kg/m^2^), normal weight (18.5–24.9 kg/m^2^), overweight (25.0–29.9 kg/m^2^) and obese (≥30.0 kg/m^2^). Information on parity was based on data from both MBRN and MoBa questionnaires and divided into three categories: 0, 1 or >1 previous birth. Maternal education was categorized as <13, 13–16, >16 years. Smoking during pregnancy was categorized as non-smoking, occasional, or daily smoking reported in Q1. Alcohol intake calculated from the FFQ was converted to a binary variable (yes or no). Iodine intake was ranked into quintiles. Dietary fibre, total protein intake and total energy intake in kilo Joules (kJ) were added as continuous variables.

### 2.6. Statistical Methods

All statistical analyses were performed using IBM SPSS Statistics version 27.0 and R version 3.5.0. All p-values were two-sided, and the thresholds for significance were corrected for the total number of regression models in the main analyses (0.05 / (4 exposure × 3 outcomes) ≈ 0.004). Differences in selenium intake and selenium status according to maternal characteristics were studied with the Kruskal–Wallis test. Prior to analysis, selenium intake was standardized, and selenium blood concentration was log transformed. Linear regression was used to analyse the association between standardized selenium intake or log transformed whole blood selenium concentration and birth weight as a continuous variable. The association between standardized selenium intake (from diet, or from supplements—organic or inorganic) or log transformed whole blood selenium concentration and SGA was analysed with logistic regression. All analyses were performed with and without adjustment for the predefined potential confounders described above.

## 3. Results

### 3.1. Selenium Intake and Status in the Study Population

In the full study sample, the median (25th–75th percentile) total intake of selenium from diet and selenium-containing supplements was 61 (48–86) µg/day. Selenium intake from diet was 53 (44–62) µg/day, and 29% had selenium intake from food above the recommended daily intake (RDI) of 60 µg/day [45]. In selenium supplement users (*n* = 23,336, 32.5% of the total population), selenium intake from supplements was 50 (30–75) µg/day and from diet was 53 (44–62) µg/day in this group. A total of 20,745 women took supplements containing inorganic supplements, 3,277 women took supplements containing organic selenium and 686 women took both inorganic and organic selenium supplements. Selenium supplement users had a slightly higher intake of selenium from diet than non-users and slightly higher blood selenium concentration (Table 1). The median (25th–75th percentile) concentration of blood selenium in the subgroup of 2572 women was 102 (89–117) µg/L, and 54% had a concentration above the reference value for adequate selenium status of 100 μg/L for whole blood [46]. The distribution of selenium intake from diet is shown in Figure 2. The most important contributors to selenium intake were cereals and grains, seafood, meat and egg, milk and dairy products. In Norway, no food items are fortified with selenium.

### 3.2. Maternal Characteristics and Selenium Intake

Table 1 describes the maternal characteristics of all 71,728 women by selenium intake and selenium status. Higher dietary intake was associated with higher age and education, lower BMI, no maternal smoking or passive smoking and no nausea during second trimester. Higher maternal dietary selenium intake was also associated with higher maternal intake of iodine, long chain n-3 polyunsaturated fatty acids, fibre, protein and energy (Table 1).

### 3.3. Birth Weight

The mean birth weight was 3582 g (standard deviation (SD): 560 g). In total, 4952 (6.9%) infants were born SGA according to the population-based growth curve. Based on ultrasound-based and customized growth curves, 1422 (2.0%) and 10,237 (14.5%) infants, respectively, were classified as SGA.

### 3.4. Maternal Dietary Selenium Intake and Birth Weight and SGA

Maternal dietary selenium intake during the first four to five months of pregnancy was significantly associated with higher birth weight and population-based birth weight z-scores adjusted *ß* = 0.027 (95% CI: 0.007, 0.014) per SD, i.e., 14.3 µg/day of selenium intake (Table 2). Likewise, significant positive associations were found for maternal selenium intake and birth weight percentiles according to ultrasound-based and customized growth standards, adjusted ß (95% CI) = 0.7 (0.3, 1.1) per SD of selenium intake for both definitions (Appendix A).

We tested for interactions between selenium intake from food and blood selenium as well as interactions between selenium intake from supplements and blood selenium on population-based z-scores of birth weight adjusted for gestational duration and sex. No interactions were found (*p*-value > 0.72), so we only present the effect estimates for selenium intake and blood selenium without the interaction.

Higher maternal dietary selenium intake was significantly associated with a lower risk of SGA according to the population-based z-score definition (adjusted OR = 0.91, 95% CI 0.86, 0.97, per SD of selenium intake) (Table 3). The results were comparable for the customized defined SGA, but not significant for the ultrasound-based SGA which included fewer SGA cases, OR (95% CI) 0.95 (0.91, 0.99) and 0.93 (0.84, 1.04) per SD of selenium intake, respectively (Appendix A).

### 3.5. Maternal Selenium Intake from Supplements and Birth Weight and SGA

The association between maternal selenium intake from supplements and birth weight was explored for organic and inorganic selenium separately (Table 4). We found no association between maternal organic supplement intake and birth weight (adjusted *ß*: −1.20 per SD, i.e., 10.4 µg/day, 95% CI: −4.40, 2.00). However, the association between maternal inorganic supplement intake and birth weight was nominally significant (adjusted *ß*: 4.30 per SD, i.e., 33.0 µg/day, 95% CI: 1.06, 7.53), but did not reach significance after Bonferroni correction for multiple comparisons. We observed similar results for both organic and inorganic selenium supplement intake when using birth weight percentiles according to ultrasound-based and customized growth standards (Appendix A). After Bonferroni correction, only the association between inorganic selenium intake and customized birth weight standards was significant in the adjusted model (*ß*: 0.83 per SD, 95% CI: 0.34, 1.32, Appendix A).

Maternal selenium intake from supplements was not associated with SGA (Table 5), adjusted OR: 1.00 per SD (95% CI: 0.97, 1.03) and OR: 1.00 per SD (95% CI: 0.97, 1.03), respectively, for organic and inorganic selenium intake from supplements. Analyses using the customized-based and ultrasound-based definitions supported these null findings (Appendix A).

### 3.6. Maternal Whole Blood Selenium Concentration, Birth Weight and Small for Gestational Age

Maternal whole blood selenium concentration was not associated with birth weight or birth weight z-scores (Table 6). Similarly, associations between whole blood selenium concentration and birth weight percentiles according to ultrasound-based and customized z-scores were non-significant (Appendix A).

Maternal whole blood selenium concentration was not associated with SGA (Table 7). Similarly, the associations between maternal whole blood selenium concentration and SGA according to the other two growth curves were also non-significant (Appendix A).

## 4. Discussion

In this study including 71,728 pregnant women, maternal dietary intake of selenium during the first half of pregnancy was weakly associated with increased birth weight (12 g per SD of selenium intake in the adjusted models) and with decreased risk of the child being born SGA (9% decrease per SD of selenium intake). However, maternal selenium intake from supplements in 71,728 mothers and whole blood selenium in 2628 mothers were not associated with birth weight or with SGA.

Foetal growth is dependent on nutrients transported from the maternal to the foetal circulation across the placenta. The transport of small membrane permeable molecules such as oxygen and carbon dioxide is influenced mainly by umbilical blood flow and placental structure, while larger molecules such as amino acids, fatty acids and glucose are dependent on nutrient transport proteins [47]. The nutrient transport capacity of the placenta is influenced by numerous factors, including hormones, nutrient levels and placental function [48]. Furthermore, oxidative stress in the placenta has been shown to influence the transport of nutrients through altering the gene expression of different nutrient transporters (e.g., glucose and amino acid) [49,50]. In vitro studies have shown that selenium supplementation protects placental cells from oxidative stress through increased expression of selenium-containing antioxidants, such as glutathione and thioredoxin reductase [51]. Hence, one of the leading hypotheses regarding how selenium may affect foetal growth is through the selenium-dependent antioxidative defence system [49,50,51,52].

Other selenium-dependent proteins are the iodothyronine deiodinase (DIOs) that are involved in thyroid hormones metabolism [53]. Thyroid hormones are essential in regulating placental nutrient transport, for example, hyperthyroidism is known to reduce circulating glucose in foetal tissues [54]. Hence, another hypothesis on how selenium may influence foetal growth is through regulating the levels of thyroid hormones. In line with this, mice fed a diet low in selenium had increased levels of both maternal and foetal plasma levels of the thyroid hormones triiodothyronine (T3) and tetraiodothyronine (T4) [49]. In humans, studies have found lipid peroxidation and oxidative stress to be associated with children being born SGA [55,56]. In one of the studies, SGA born children had lower levels of reduced glutathione (indicating deficient antioxidant defence mechanisms) but higher concentration of the lipid peroxidation malondialdehyde [56].

Previous observational studies analysing the association between selenium status and birth weight or SGA births have mainly focused on selenium concentration in maternal serum, plasma or blood samples, collected either before or during pregnancy [22,23,27] or at delivery [57,58,59,60,61]. To the best of our knowledge, no study has investigated the association between maternal dietary intake of selenium and birth weight. Previous studies have suggested that maternal selenium status during pregnancy, but not at delivery, correlates positively with birth weight. A nested case-control study within the US Camden Study, with preterm delivery as the main outcome, found selenium status at around 16 weeks of gestation to be positively associated with birth weight in infants born at term (*n* = 126), but not in infants born preterm (*n* = 107) [27]. Another study performed in Japan on 44 newborn infants, including maternal serum samples collected around entry to antenatal care, found maternal selenium status during pregnancy to be positively associated with higher birth weight [23]. A UK study on 126 adolescent pregnancies found selenium concentration in plasma collected at around 30 weeks of gestation to be associated with higher birth weight scores and with a lower risk of SGA [22]. These suggested positive associations between maternal selenium status and birth weight were not supported in the sub-sample in the current study, where selenium status was not associated with birth weight or SGA. We had information about selenium status in a fairly small subgroup of the women (3.7%). Still, the current study is far larger than previous studies. One obvious difference between our study and the previous studies is that we used selenium analysed in whole blood, while the three other studies used selenium analysed in serum or plasma. Whole blood selenium concentration reflects both status and uptake while plasma/serum selenium only reflects short-term status [46]. Hence, the difference in specimen used for selenium analyses may be one explanation for the different results between the current and previous studies. In the current study, median selenium status was close to the reference level of 100 µg/L [46,62].

Possible reasons for why only selenium intake from diet but not from supplements was associated with birth weight and SGA in this study can only be speculated on. One reason may be that the women in this cohort have intake levels from diet close to the level associated with optimal selenoprotein expression and adding further selenium intake from supplements does not give further improvement in selenoprotein expression. Another reason might be related to the difficulty in accurately measuring the intake of nutrients contributed from dietary supplements. Dietary intakes are often more stable over time while intake from supplements may vary more, especially during pregnancy when many women care more about their nutritional intake [63,64]. One additional explanation is that dietary selenium intake may serve as a proxy for a high quality diet containing other factors that have a beneficial effect on birth weight. As we state in the introduction, for example, high sugar consumption [14], low fish intake [15,16], low iodine intake [17], and high caffeine consumption [18,19] have previously been associated with the risk of being born SGA. However, we adjusted our models for fibre intake as a proxy for an overall healthy diet, iodine intake, protein intake and n-3 intake from diet. After adjusting for these and other factors, the effect size was somewhat reduced for selenium intake association with birth weight but not for SGA. Hence, even if we cannot rule out that dietary intake of selenium is a proxy for another dietary factor that correlates with the intake of selenium, we believe that the size of our study, allowing us to adjust for very many factors including markers for overall food quality, strengthens the validity of our results. A possible explanation for the lack of association between blood selenium and birth weight may be that the sub-population with available blood selenium was rather homogenous and had fairly good selenium status.

The main strength of this study is its large sample size. The study is by far the largest examining selenium intake from both diet and supplements and whole blood selenium concentration in relation to birth weight and SGA. Another strength of the current study is its prospective design with selenium intake assessment during the first half of pregnancy and with maternal blood collected in mid-pregnancy, since previous studies suggest that maternal selenium status during pregnancy, but not at delivery, is associated with birth weight [22,23,27,57,58,59,60,61].

Another strength is the population-based study design, including women from all over Norway representing women living in different areas and regions, from different socioeconomic groups and women with diverse dietary habits. Another strength is the extensive data from questionnaires and the linkage to MBRN allowing us to control for important covariates.

Still, our study is limited by the observational design and the inherent risk of residual confounding. There is also a potential influence of self-selection bias as for all cohort studies. The participation rate in MoBa was 41%; women participating in MoBa are generally healthier and better educated than the general population of pregnant Norwegian women [28].

In the main analyses, we used the birth weight z-score variable that was extracted directly from MBRN. To make the results more comparable to those from other parts of the world, we analysed z-scores of birth weight and SGA according to different growth standards. The findings for the association between maternal dietary selenium intake during pregnancy and higher birth weight and lower risk for SGA were consistent and of similar size when using the different growth standards as shown in the(Appendix A).

The subgroup of women with data on selenium concentration in whole blood was included in the Norwegian Environmental Biobank Study, where two of the inclusion criteria for participation was to have answered the first six MoBa questionnaires and to have provided biological samples. Hence, this subgroup with selenium measurements does not represent the whole MoBa population, but a highly selected group. The women in this subgroup included a higher proportion of non-smokers and highly educated compared with those in the whole MoBa [34]. Thus, a limitation with the current study is that the generalizability of the selenium blood status may be compromised due to selection bias. In addition, while we did not identify any association between whole blood selenium and birth weight or SGA, we cannot rule out that smaller effect estimates might be detected if a larger and more varied study population with broader variation in selenium concentration was available.

The dietary intake of selenium was estimated during pregnancy using an FFQ specifically developed and validated for use in the MoBa cohort [29,31]. All dietary assessment methods have errors. The use of an FFQ to assess diet has limitations since it is imprecise and prone to misreporting. Since the selenium content varies according to the selenium concentration of the soil where crops are grown or the animals graze, it is especially difficult to assess selenium intake from food composition databases. This difficulty is reflected in the low correlation between selenium intake from diet and selenium concentration in blood reported previously (Spearman rho: 0.135 95% CI: 0.10, 0.17) [26]. However, blood selenium concentration is influenced by homeostatic regulation and by individual differences in absorption, metabolism and body composition. In the validation study, selenium intake estimated with the FFQ was significantly correlated with selenium intake estimated using a 4-day food diary, but the correlation coefficient was low (Spearman’s rho 0.33 (95%CI: 0.16, 0.48)) [32].

## 5. Conclusions

A maternal diet rich in selenium during pregnancy may be beneficial for infant birth weight and for reducing the risk of SGA births. These results were not supported by findings for selenium intake from supplements. Around half of the women had selenium intake levels below the recommended 60 µg/day and around half had selenium status below the reference value of 100 µg/L. Hence, the association seen between selenium intake and birth weight and SGA may not be generalisable to populations with other levels of selenium intake and status. In addition, due to the observational design of this study, residual confounding may exist. Further experimental studies are needed to establish causality.

## Figures and Tables

**Figure 1 nutrients-13-00023-f001:**
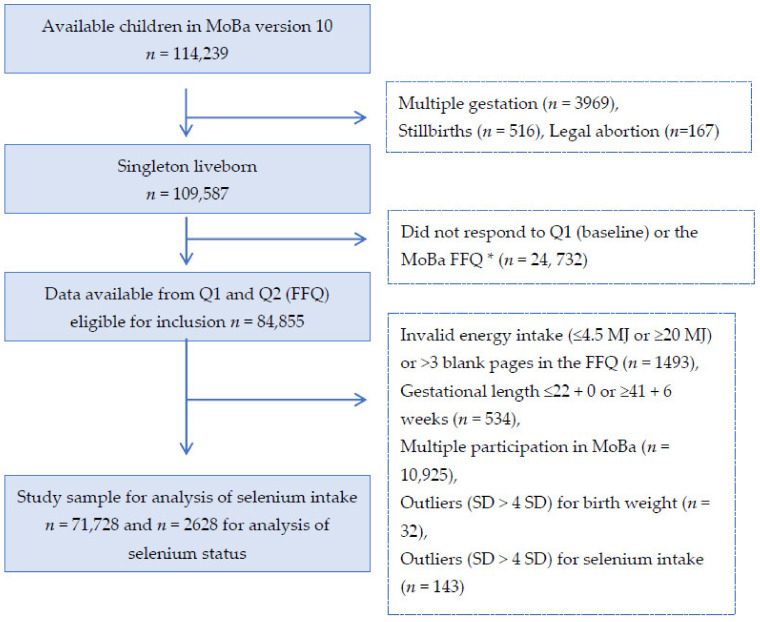
Flow chart over the study population. * The current food frequency questionnaire (FFQ) was not used before 2002, explaining the large drop of numbers from box 2 to 3.

**Figure 2 nutrients-13-00023-f002:**
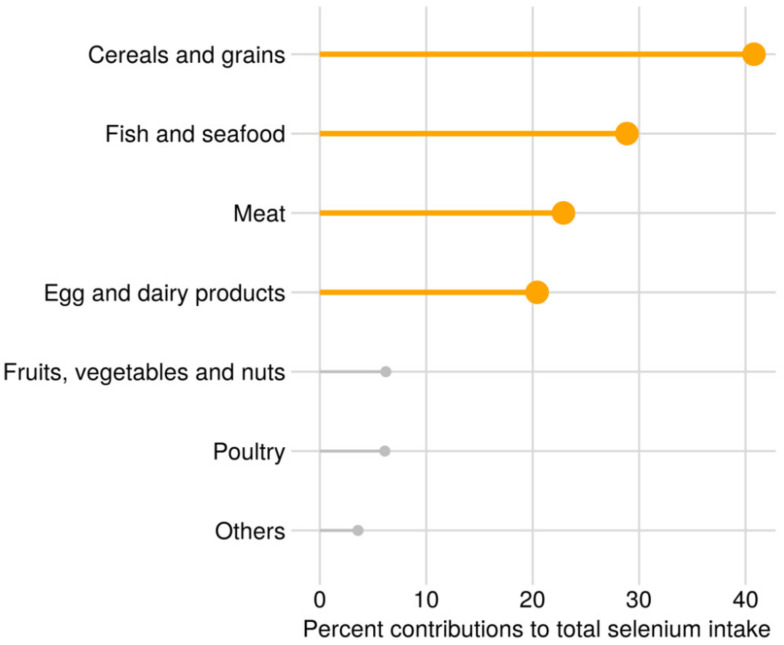
Sources of selenium from food, percent mean contribution to the total dietary selenium intake.

**Table 1 nutrients-13-00023-t001:** Selenium intake and selenium status by maternal characteristics.

		Selenium from Diet µg/day			Selenium in Whole Blood µg/L		
		*n*	Median	25th–75th Percentile	*n*	Median	25th–75th Percentile
Total population		71,728	53	44–62	2,628	102	89–117
Maternal age, years	<25	8239	50	41–61	220	98	86–112
	25–29	24,313	52	43–62	945	102	88–116
	30–34	26,884	53	45–63	1173	102	90–118
	>34	12,292	54	46–64	290	104	92–120
Pre-pregnancy BMI, kg/m^2^	<18.5	2130	53	44–64	81	104	91–119
	18.5–24.9	45,976	53	45–63	1699	103	90–119
	25–29.9	15,134	52	43–61	608	101	88–115
	>30	6633	51	42–61	193	100	86–114
	Missing	1855	52	44–62	47	100	93–111
Parity	0	38,169	52	43–62	1511	104	91–119
	1	21,557	53	44–62	751	101	88–115
	2+	11,945	54	45–63	365	99	87–114
	Missing	57	56	45–63	1	112	112–112
Maternal education, years	<13	22,286	51	42–62	674	100	86–114
	13–16	29,757	53	44–62	1244	101	88–116
	>16	18,150	54	46–64	653	107	94–122
	Missing	1535	51	42–62	57	99	87–115
Maternal smoking during pregnancy	Never	65,504	53	44–62	2452	102	90–118
	Occasionally	1943	52	43–63	63	99	91–113
	Daily	3876	51	42–62	103	95	85–110
	Missing	405	52	43–62	10	97	93–108
Passive smoking	No	62,767	53	44–62	2364	102	89–117
	Yes	7580	52	43–63	224	100	87–114
	Missing	1381	51	43–63	40	101	86–119
Nausea during second trimester	No	63,504	53	44–62	2341	102	90–118
	Yes	8224	52	42–62	287	97	87–114
Gestational age, weeks	<33	942	51	43–61	7	90	83–96
	34–36	2654	53	44–62	72	104	93–114
	37–38	11,612	53	44–62	408	102	89–117
	>38	56,520	53	44–62	2141	102	89–117
Iodine intake, µg/day	<89	17,942	43	36–50	641	101	88–118
	89–121	17,942	50	43–58	676	104	89–118
	121–162	17,934	55	48–63	685	102	90–118
	>162	17,831	63	55–73	626	101	89–116
Long-chain n-3 PUFA intake from diet, g/day	≤0.24	23,934	45	44–68	866	98	87–112
	0.24–0.43	23,911	52	45–61	963	104	90–118
	≥0.43	23,804	61	52–71	799	106	92–120
Fibre intake, g/day	<26	23,951	43	37–51	867	101	88–116
	26–34	23,941	53	46–60	931	102	89–117
	>34	23,836	63	55–72	830	104	90–118
Protein intake, g/day	<76	23,952	42	36–48	864	102	88–117
	76–93	23,950	53	47–59	905	102	90–117
	>93	23,826	65	58–74	859	102	90–117
Energy intake, MJ/day	<8389	23,950	44	37–50	909	103	90–118
	8392–10,460	23,946	53	46–60	892	102	89–117
	>10,460	23,832	63	55–73	827	101	89–116
Selenium supplement intake	Yes	23,336	53	44–63	814	100	88–115
	No	48,392	52	44–62	1814	106	94–121

Amount of daily selenium intake from food (FFQ data) and concentration of blood selenium in mid pregnancy according to maternal characteristics, from 71,728 participants in the Norwegian Mother and Child Cohort Study. Selenium intake from food was assessed with a food frequency questionnaire in gestational week 22. Blood selenium concentration was measured in whole blood collected in gestational week 17–18 in a subsample of 2628. Abbreviations: BMI—Body Mass Index; MJ—Mega Joule; PUFA—polyunsaturated fatty acids.

**Table 2 nutrients-13-00023-t002:** Associations between maternal dietary selenium intake and birth weight in 71,728 women from the Norwegian Mother, Father and Child Cohort study.

		Birth Weight, Grams			Z-Scores		
	*n*	*ß*^2^ (95% CI)	SE	P	*ß*^2^ (95% CI)	SE	P
Unadjusted	71,728	18 (14, 22)	2.1	2.2 × 10^−17^	0.032 (0.025, 0.039)	0.004	6.2 × 10^−19^
Adjusted ^1^	66,923	12 (6, 18)	3.1	1.0 × 10^−4^	0.027 (0.013, 0.040)	0.007	1.0 × 10^−4^

Multiple linear regression analysis of standardized selenium intake from food in relation to birth weight in grams and z-scores of birth weight according to gestational age. Birth weight z-scores were calculated according to population-based growth curves. ^1^ Adjusted for maternal age at delivery, maternal pre-pregnancy body mass index (BMI), parity, maternal smoking during pregnancy, passive smoking, nausea during second trimester, maternal education, fibre intake, iodine intake, protein intake, n-3 intake from diet, selenium intake from supplements and total energy intake. ^2^
*ß* per SD of selenium intake in µg/day, i.e., 14.3 µg/day. Unadjusted models included 71,728 women while the adjusted models included 66,923 women due to missing data on the covariates. Abbreviations: *ß*—beta, CI—confidence interval, SE—standard error.

**Table 3 nutrients-13-00023-t003:** Associations between maternal dietary selenium intake and small for gestational age (SGA) birth in 71,728 women from the Norwegian Mother, Father and Child Cohort study.

			SGA	
	*n* total	*n* SGA	OR ^2^ (95% CI)	*p*
Unadjusted	71,728	4952	0.92 (0.90, 0.95)	8.24 × 10^−8^
Adjusted ^1^	66,923	4614	0.92 (0.86, 0.97)	0.003

Multiple logistic regression of standardized maternal dietary selenium intake in relation to small for gestational age according to population-based growth curve. ^1^ Adjusted for maternal age at delivery, maternal pre-pregnancy body mass index (BMI), parity, maternal smoking during pregnancy, passive smoking, nausea during second trimester, maternal education, fibre intake, iodine intake, protein intake, n-3 intake from diet, selenium intake from supplements and total energy intake. ^2^ OR per SD of selenium intake in µg/day, i.e., 14.3 µg/day. Unadjusted models included 71,728 women while the adjusted model included 66,923 women due to missing data on the covariates. Abbreviations: CI—confidence interval, OR—odds ratio, SGA—small for gestational age.

**Table 4 nutrients-13-00023-t004:** Associations between maternal selenium intake from supplements and birth weight in 71,728 women from the Norwegian Mother, Father and Child Cohort study.

			Birth Weight, Grams			Z-Scores		
		*n*	*ß*^2^ (95% CI)	SE	*p*	*ß*^2^ (95% CI)	SE	*p*
Organic selenium	Unadj	71,698	−0.79 (−4.89, 3.30)	2.09	0.705	−0.01 (−0.01, 0.00)	0.00	0.140
	Adj ^1^	65,772	−1.20 (−4.40, 2.00)	1.63	0.462	0.00 (−0.01, 0.00)	0.00	0.512
Inorganic selenium	Unadj	71,698	−5.08 (−9.18, −0.98)	2.09	0.015	−0.01 (−0.01, 0.00)	0.00	0.033
	Adj ^1^	65,772	4.30 (1.06, 7.53)	4.30	0.009	0.02 (0.01, 0.00)	0.01	0.024

Multiple linear regression analysis of standardized selenium intake from supplements in relation to birth weight in grams and z-scores of birth weight according to gestational age. Birth weight z-scores were calculated according to population-based growth curves. ^1^ Adjusted for food selenium intake, maternal age at delivery, maternal pre-pregnancy body mass index (BMI), parity, maternal smoking during pregnancy, passive smoking, nausea during second trimester, maternal education, fibre intake, iodine intake, protein intake, n-3 intake from diet, selenium intake from supplements (inorganic and organic for its respective model) and total energy intake.^2^
*ß* per SD of selenium intake from supplements in µg/day. SD for organic selenium: 10.4 µg/day and for inorganic selenium: 33.0 µg/day. Unadjusted models included 71,698 women while the adjusted models included 65,772 women due to missing data on the covariates. Abbreviations: Adj—adjusted, *ß*—beta, CI—confidence interval, SE—standard error, Unadj—unadjusted.

**Table 5 nutrients-13-00023-t005:** Associations between maternal selenium intake from supplements and small for gestational age status in 71,728 women from the Norwegian Mother, Father and Child Cohort study.

				SGA	
		*n* Total	*n* SGA	OR ^2^ (95% CI)	*p*
Organic selenium	Unadjusted	71,723	4952	1.01 (0.98, 1.03)	0.601
	Adjusted ^1^	66,923	4614	1.00 (0.97, 1.03)	0.980
Inorganic selenium	Unadjusted	71,723	4952	1.02 (0.99, 1.04)	0.256
	Adjusted ^1^	66,923	4614	0.99 (0.96, 1.02)	0.588

Multiple logistic regression of standardized maternal intake of selenium from supplements in relation to small for gestational age according to population-based growth curve. ^1^ Adjusted for dietary selenium intake, maternal age at delivery, maternal pre-pregnancy body mass index (BMI), parity, maternal smoking during pregnancy, passive smoking, nausea during second trimester, maternal education, fibre intake, iodine intake, protein intake, n-3 intake from diet, selenium intake from supplements (inorganic and organic for its respective model) and total energy intake. ^2^ OR per SD of selenium intake from supplements in µg/day. SD for organic selenium: 10.4 µg/day and for inorganic selenium: 33.0 µg/day. Unadjusted models included 71,723 women while the adjusted model included 66,923 women due to missing data on the covariates. Abbreviations: CI—confidence interval, OR—odds ratio, SGA—small for gestational age.

**Table 6 nutrients-13-00023-t006:** Associations between maternal selenium concentration in whole blood and birth weight in 2628 women from the Norwegian Mother, Father and Child Cohort study.

		Birth Weight, Grams			Z-Scores		
	*n*	*ß*^2^ (95% CI)	SE	*p*	*ß*^2^ (95% CI)	SE	*p*
Unadjusted	2628	−0.97 (−1.89, −0.05)	46.8	0.038	−0.003 (−0.004, −0.001)	0.09	0.005
Adjusted ^1^	2488	−0.47 (−1.29, 0.34)	41.6	0.254	−0.001 (−0.003, 0.001)	0.09	0.195

Multiple linear regression analysis of log transformed maternal selenium concentration in whole blood collected during pregnancy in relation to birth weight in grams, z-scores of birth weight according to gestational age. Birth weight z-scores were calculated according to population-based growth curve. ^1^ Adjusted for maternal age at delivery, maternal pre-pregnancy body mass index (BMI), parity, maternal smoking during pregnancy, passive smoking and maternal education. ^2^
*ß* per % of blood selenium. Crude models included 2628 women while adjusted models included 2488 women due to missing data on the covariates. Abbreviations: *ß*—beta, CI—confidence interval, SE—standard error.

**Table 7 nutrients-13-00023-t007:** Associations between maternal selenium concentration in whole blood and small for gestational age status in 2628 women from the Norwegian Mother, Father and Child Cohort study.

			SGA	
	*n* Total	*n* SGA	OR^2^ (95% CI)	*p*
Unadjusted	2628	146	1.70 (0.77, 3.70)	0.185
Adjusted^1^	2488	136	1.24 (0.53, 2.88)	0.623

Multiple logistic regression of log transformed maternal selenium concentration in whole blood collected during pregnancy in relation to small for gestational age according to population-based growth curves. ^1^ Adjusted for maternal age at delivery, maternal pre-pregnancy body mass index (BMI), parity, maternal smoking during pregnancy, passive smoking and maternal education. ^2^ OR per % of blood selenium. Crude models included 2628 women while adjusted models included 2488 women due to missing data on the covariates. Abbreviations: CI—confidence interval, OR—odds ratio, SGA—small for gestational age.

## Data Availability

Data sharing is not applicable to this article.

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
