# Peer review of "Maternal Dietary Selenium Intake during Pregnancy Is Associated with Higher Birth Weight and Lower Risk of Small for Gestational Age Births in the Norwegian Mother, Father and Child Cohort Study"

_nutrients, 2020, doi:10.3390/nu13010023_

Round 1
Reviewer 1 Report
The manuscript entitled “ Maternal Dietary Selenium Intake during Pregnancy 2 is Associated with Higher Birth Weight and Lower Risk 3 of Small for Gestational Age Births in the Norwegian 4 Mother, Father and Child Cohort Study” is a well written and highly interesting study that appears to be the largest analysis comparing selenium intake, status and birth weight. The study has been well performed and there is a very large amount of new knowledge contained within. In particular, the subanalysis of supplements based on organic/inorganic. I have a number of comments below for the authors to address
- The authors discus the potential impact of selenium deficiency in pregnancy on birth weight and programmed disease in offspring but do not cite key papers in this field. Several animal studies have demonstrated these outcomes. Please cite appropriate animal studies supporting such links.
- Selenium concentrations in food products strongly depends on the selenium concentration in soil or animal feed products. How reliable therefore are food frequency questionnaires for determining selenium intake. Can you please comment on how accurately the selenium intake values match serum selenium status?
- I was also wondering why women with babies born 4 standard deviations from mean birth weight were excluded. I would have thought the most extreme birth weighs would have specific pregnancy complications that may in fact be related to selenium deficiency such as GDM which frequently increases birth weight. Do you lose the association when you include these due to low levels of selenium increasing birth weight in this sub-cohort of women. Can you comment of this factor in you study?
- Why was the selenium intake only assessed during the first half of pregnancy if birth weight at the end of pregnancy was your major question? This is particularly of relevance given that a previous study suggested that selenium status at term was not associated with birth weight and it would have been really interesting to confirm this in your large study.
- It was interesting that selenium intake was adjusted for fish, meat and dairy. Other food products are known to contain much higher levels of selenium (such as Brazil nuts) or have a similar amount of selenium but are eaten in much larger quantities (wheat).
- Regardless, selenium content in food depends on soil or animal feed intake. Given the use of selenium fortified pastures for meat production in some countries in northern Europe (such as Finland), do you have any information about the country of origin of your meat products in this study and have you factored this into your analysis?
- It was quite surprising that as many as 29% had selenium intake values above recommended levels. Indeed, this population study seems to show that selenium concentrations in Norwegian women are much more closely aligned with recommended concentrations (both intake and blood concentration) than studies in other countries such as the UK or Australia. Do you speculate that this is due to the high selenium concentration in soil in Norway compared to other nations or improved dietary behaviours in pregnant women? Would you also comment on the possibility that in other countries, the associations may be different given that fewer women would be in such a healthy range and that there may be more extreme selenium differences in those other regions.
- You speculate that associations between selenium status and birth weight may be due to antioxidant activity. However, you also highlight the important role of selenium for deiodinase activity in your introduction but do not comment on this in the discussion. Given the growing evidence that selenium deficiency may contribute to low birth weight through altering thyroid health, can you please provide discussion on this point in the discussion as an alternative possible cause.
Author Response
Reviewer 1
The manuscript entitled “ Maternal Dietary Selenium Intake during Pregnancy is Associated with Higher Birth Weight and Lower Risk of Small for Gestational Age Births in the Norwegian 4 Mother, Father and Child Cohort Study” is a well written and highly interesting study that appears to be the largest analysis comparing selenium intake, status and birth weight. The study has been well performed and there is a very large amount of new knowledge contained within. In particular, the subanalysis of supplements based on organic/inorganic. I have a number of comments below for the authors to address
ANSWER: We want to thank the reviewer for the valuable comments on our manuscript. All changes have been highlighted in the revised paper, and detailed response to all comments follows below.
Reviewer 1 comment #1: The authors discus the potential impact of selenium deficiency in pregnancy on birth weight and programmed disease in offspring but do not cite key papers in this field. Several animal studies have demonstrated these outcomes. Please cite appropriate animal studies supporting such links.
ANSWER: We agree that this should have been included in the paper. Also based on your last comment we have now included this section into the discussion, page…, line…:
“Foetal growth is dependent on nutrients transported from the maternal to the fetal circulation across the placenta. The transport of small membrane permeable molecules such as oxygen and carbon dioxide is influenced mainly by umbilical blood flow and placental structure while larger molecules such as amino acids, fatty acids and glucose are dependent on nutrient transport proteins {Gaccioli, 2016 #2642}. The nutrient transport capacity of the placenta is influenced by numerous factors including hormones, nutrient levels and placental function {Lager, 2012 #2643}. Furthermore, oxidative stress in the placenta has been shown to influence the transport of nutrients through altering the gene expression of different nutrient transporters (e.g., glucose and amino acid) {Hofstee, 2019 #2635;Araújo, 2013 #2640}. In vitro studies have shown that selenium supplementation protects placental cells from oxidative stress through increased expression of selenium-containing antioxidants, such as glutathione and thioredoxin reductase {Khera, 2013 #2637}. Hence, one of the leading hypothesis regarding how selenium may affect foetal growth is through the selenium-dependent antioxidative defence system {Hofstee, 2019 #2635;Araújo, 2013 #2640}{Hofstee, 2019 #2635;Khera, 2015 #2638;Khera, 2013 #2637}.
Other selenium-dependent proteins are the iodothyronine deiodinase (DIOs) that are involved in thyroid hormones metabolism {Brown, 2007 #2647}. Thyroid hormones are essential in regulating placental nutrient transport, for example, hyperthyroidism is known to reduce circulating glucose in foetal tissues {Boelen, 2009 #2641}. Hence, another hypothesis on how selenium may influence foetal growth is through regulating the levels of thyroid hormones. In line with this, mice fed a diet low in selenium, had increased levels of both maternal and fetal plasma levels of the thyroid hormones triiodothyronine (T3) and tetraiodothyronine (T4) {Hofstee, 2019 #2635}.”
Reviewer 1 comment #2: Selenium concentrations in food products strongly depends on the selenium concentration in soil or animal feed products. How reliable therefore are food frequency questionnaires for determining selenium intake. Can you please comment on how accurately the selenium intake values match serum selenium status?
ANSWER: In our first publication on selenium using data from the same cohort (MoBa), we reported the correlations between selenium intake from diet and supplements and blood selenium blood concentration. The correlation between selenium from diet and blood selenium blood was 0.135 (95% CI: 0.10, 0.17), which is a weak correlation but does not necessarily infer that the estimated selenium intake is inaccurate. Biomarkers in blood reflect the overall status and are generally weakly associated with intake due to homeostatic control and other factors. In response to the reviewer’s comment, we have added the following sentence to the last part of the discussion where we discuss the difficulties in estimating selenium intake accurately, page…, line…:
“This difficulty is reflected in the low correlation between selenium intake from diet and selenium concentration in blood reported previously (Spearman rho: 0.135 95% CI: 0.10, 0.17) {Barman et al. 2019}. However, blood selenium concentration is influenced by homeostatic regulation and by individual differences in absorption, metabolism and body composition.”
Reviewer 1 comment #3: I was also wondering why women with babies born 4 standard deviations from mean birth weight were excluded. I would have thought the most extreme birth weighs would have specific pregnancy complications that may in fact be related to selenium deficiency such as GDM which frequently increases birth weight. Do you lose the association when you include these due to low levels of selenium increasing birth weight in this sub-cohort of women. Can you comment of this factor in you study?
ANSWER: The data on birth weight contained some extreme values and outliers. We needed to clean the data before using it since the estimates from linear regressions are affected by outliers. In total we excluded 32 subjects only with >4SD of birth weight (two of them zscore > 8 SD).
We ran the analyses including these subjects in the model and obtained the same results: For birth weight (grams) and birth weight zscores, the estimates were 12.0 (95% CI: 5.8, 18.1; p-value- 1.4*10-4) and 0.03 (95% CI: 0.01, 0.04; p-value= 1.5* 10-4) after adjusting for the same confounders as in the manuscript. For SGA, OR 0.9 (95% CI: 0.86, 0.97; pvalue= 0.003), also for the adjusted model.
Reviewer 1 comment #4: Why was the selenium intake only assessed during the first half of pregnancy if birth weight at the end of pregnancy was your major question? This is particularly of relevance given that a previous study suggested that selenium status at term was not associated with birth weight and it would have been really interesting to confirm this in your large study.
ANSWER: We agree that it would have been interesting to have had selenium measurements at several time points during pregnancy. However, in the MoBa, data on selenium intake from food was only obtained for the first half of pregnancy and blood samples were donated once in gestational week 18. However, diet during pregnancy changes little throughout gestation (e.g. Rifas-Shiman 2006) and therefore it is unlikely that selenium intake would differ substantially if assessed at a later stage.
Reviewer 1 comment #5: It was interesting that selenium intake was adjusted for fish, meat and dairy. Other food products are known to contain much higher levels of selenium (such as Brazil nuts) or have a similar amount of selenium but are eaten in much larger quantities (wheat).
ANSWER: These variables were defined as confounders as they have been shown in previous studies to be associated with birth weight, possibly due to other compounds than selenium. Since they also contribute to selenium intake, we adjusted for these variables (i.e., fish and total protein). No such association is known for wheat intake. Regarding Brazil nuts, consumption is extremely low in Norway and the MoBa FFQ did not assess intake of this. As shown in Figure 2, grain (of which wheat is dominant) is the primary source of dietary selenium. Till around 1990, wheat was imported from selenium-rich areas in the US and Canada, but was then replaced by locally grown wheat and import from Europe and the Middle East, with much lower selenium due to low soil selenium. The proportions of self-grown and imported grain were relatively stable in the period when participants were recruited to MoBa. Norway does not fortify the flour with selenium, and the selenium concentration in flour used in the food composition table reflects the average over the study years.
Reviewer 1 comment #6: Regardless, selenium content in food depends on soil or animal feed intake. Given the use of selenium fortified pastures for meat production in some countries in northern Europe (such as Finland), do you have any information about the country of origin of your meat products in this study and have you factored this into your analysis?
ANSWER: Norway has a high degree of self-sufficiency of meat production (more than 90%) and pastures are not fortified so this issue is not relevant and has not been considered in our analyses.
Reviewer 1 comment #7: It was quite surprising that as many as 29% had selenium intake values above recommended levels. Indeed, this population study seems to show that selenium concentrations in Norwegian women are much more closely aligned with recommended concentrations (both intake and blood concentration) than studies in other countries such as the UK or Australia. Do you speculate that this is due to the high selenium concentration in soil in Norway compared to other nations or improved dietary behaviours in pregnant women? Would you also comment on the possibility that in other countries, the associations may be different given that fewer women would be in such a healthy range and that there may be more extreme selenium differences in those other regions.
ANSWER: The blood selenium concentration in Norway has declined steadily over the last thirty years due to higher use of local and European wheat and the following decline in wheat selenium. The fact that 29% of our study participants had selenium values that reached the recommended daily intake does not ensure sufficiency in the whole population. As explained in a previous comment, the selenium concentration in Norwegian soil is low, but their dietary habits with a relatively high intake of bread and seafood may explain why selenium status is higher in Norway than in other countries such as the UK or Australia.
We agree with the reviewer “that in other countries, the associations may be different given that fewer women would be in such a healthy range and that there may be more extreme selenium differences in those other regions.” This is addressed in the discussion on page 12 where we instead write “While we did not identify any association of whole blood selenium status with birth weight or SGA, we cannot rule out that significant associations might be detected if a larger and more varied study population with broader variation in selenium concentration was studied.”
Reviewer 1 comment #8 You speculate that associations between selenium status and birth weight may be due to antioxidant activity. However, you also highlight the important role of selenium for deiodinase activity in your introduction but do not comment on this in the discussion. Given the growing evidence that selenium deficiency may contribute to low birth weight through altering thyroid health, can you please provide discussion on this point in the discussion as an alternative possible cause.
ANSWER: please see answer to comment 1.
Reviewer 2 Report
This is an interesting analysis over a big size cohort. However, you should try to provide an explanation for the association of maternal selenium intake from diet with birth weight, but not for selenium supplements or selenium blood levels. Could be the dietary intake of selenium a proxy of a high quality diet? Is there some anologies for other micronutrients?
In line 207 there is a mistake for selenium unit (mg/day instead of µg/day)
Author Response
Reviewer 2
Reviewer 2 comment 1: This is an interesting analysis over a big size cohort. However, you should try to provide an explanation for the association of maternal selenium intake from diet with birth weight, but not for selenium supplements or selenium blood levels. Could be the dietary intake of selenium a proxy of a high quality diet? Is there some anologies for other micronutrients?
Answer: Thanks for pointing this out, we agree that this is a part of our results that needs to be discussed. We have added a section in the discussion. But we agree that your suggestion on selenium being a proxy for a high quality diet is valid and needs to be mentioned as an alternative hypothesis for the disagreement in result for dietary and supplement selenium. We have added the following to the manuscript, page…, line…:
“One alternative explanation for the difference in the association between dietary selenium and supplementary selenium and blood selenium levels and birth weight and SGA, is that dietary selenium intake may serve as a proxy for a high quality diet containing other factors that have a beneficial effect on birth weight. As we state in the introduction for example high sugar consumption [14], low fish intake [15,16], low iodine intake [17], and high caffeine consumption [18,19] have previously been associated with the risk of being born SGA. However, we adjusted our models for fiber intake as a proxy for an overall healthy diet, iodine intake, protein intake and n-3 intake from diet. After adjusting for these and other factors, the effect size was somewhat reduced for selenium intake association with birth weight but not for SGA. Hence, even if we cannot rule out that dietary intake of selenium is a proxy for another dietary factor that correlates with the intake of selenium, we believe that the size of our study, allowing us to adjust for very many factors including markers for overall food quality, strengthen the validity of our results.”
Reviewer 2 comment 2: In line 207 there is a mistake for selenium unit (mg/day instead of µg/day)
Answer: Thanks for noticing this error. This has now been corrected.